# Carcass and Meat Quality Traits in Female *Lidia* Cattle Slaughtered at Different Ages

**DOI:** 10.3390/ani14060850

**Published:** 2024-03-10

**Authors:** Miguel Ángel Cantarero-Aparicio, Elena Angón, Carlos González-Esquivel, Francisco Peña, Javier Caballero-Villalobos, Eoin G. Ryan, José Manuel Perea

**Affiliations:** 1Departamento de Producción Animal, Universidad de Córdoba, 14071 Cordoba, Spain; t42caapm@uco.es (M.Á.C.-A.); pa1peblf@uco.es (F.P.); javier.caballero@uco.es (J.C.-V.); jmperea@uco.es (J.M.P.); 2Instituto de Investigaciones en Ecosistemas y Sustentabilidad (IIES), Universidad Nacional Autónoma de México (UNAM), Morelia 58190, Mexico; cgesquivel@cieco.unam.mx; 3Section of Herd Health and Animal Husbandry, School of Veterinary Medicine, University College Dublin, D04 V1W8 Belfield, Ireland; eoin.g.ryan@ucd.ie

**Keywords:** *Lidia* cattle, carcass traits, meat quality, sensory panel, local breeds

## Abstract

**Simple Summary:**

The *Lidia* breed is an autochthonous Spanish breed linked to the pasture-based *dehesa* system, where extensive livestock farming is practiced and plays an important role in ecosystem conservation and rural development. This system is linked to the conservation of biodiversity and sustainable agricultural practices. However, it faces important challenges that make it necessary to explore new strategies to help farmers. In this study, we sought to understand the carcass and meat traits (technological and sensorial) as a strategy that would allow it to compete and differentiate itself from other meats. We used 300 *Lidia* females slaughtered at different ages. Age at slaughter influenced meat quality, with particular importance on sensory variables such as flavor, juiciness, overall tenderness and overall acceptability. Technological variables were acceptable and the results of a trained sensory panel pointed to the good qualities of this meat. To our knowledge, this is the first work evaluating carcass traits and the technological and sensory quality of meat with a large sample of animals of the *Lidia* breed.

**Abstract:**

The aim of this study was to assess the carcass and meat quality of female *Lidia* cattle slaughtered at different ages, in order to deepen our understanding of the breed’s unique characteristics. The effect of slaughter age on carcass traits and meat quality attributes of m. *Longissimus* was investigated in *Lidia* heifers (*n* = 200) and cows (*n* = 100) reared and finished in an extensive system. The animals were slaughtered at 24–36 months (Heifer I), 36–48 months (Heifer II) or >48 months (Cull cow). The carcasses (~120 kg) presented poor conformation (O, O+) and medium fatness (2, 2+). The dissection of the 6th rib yielded mean values of 58.6%, 14.3% and 24.8% for lean, fat and bone, respectively. The cows had a higher proportion of dissectible fat (*p* < 0.05). Subcutaneous fat was classified as dark and yellowish, and meat (aged for 21 days) as dark (L* = 25.5), reddish (a* = 14.4) and moderately yellowish (b* = 12.9), with acceptable water-holding capacity (TL = 5.34%; DL = 0.97%; PL = 8.9%; CL = 22.1%) and intermediate tenderness (WBSF = 4.6 kg/cm^2^). The b* value of meat was higher (*p* < 0.05) in cull cows. The meat of cull cows was more yellowish (*p* < 0.05) and obtained higher scores for flavor (*p* < 0.05), juiciness *p* < 0.01), overall tenderness (*p* < 0.001) and overall acceptance (*p* < 0.001).

## 1. Introduction

The *Lidia* cattle breed holds a significant position among Spanish native breeds, due both to its census and geographical distribution [1]. This breed has experienced relative geographic isolation, resulting in subpopulations or “*encastes*” with notable differentiations [2]. Pelayo et al. [3] identifies the *Lidia* breed as a racial grouping arising from selective pressures for behavioral phenotypes [4].

The farming system of the *Lidia* breed has evolved in pasture-based systems predominantly in the landscapes of Spanish *dehesas* [5]. With grasslands interspersed with oak trees (*Quercus* spp.), these areas offer an ideal environment for the small- to medium-sized *Lidia* breed. From an environmental perspective, breeding *Lidia* cattle in *dehesas* positively impacts biodiversity conservation and animal welfare, promoting sustainable agricultural practices [6]. Additionally, the presence of the *Lidia* breed in rural areas plays a pivotal role in rural development, offering an economic alternative to counteract depopulation and generating sustainable employment [7].

Traditionally reared for bullfighting by emphasizing physical and temperamental traits, the conservation of the *Lidia* breed faces the need to explore new perspectives for its potential survival. In a global context where people demand evolution and sustainability, a transition towards high-quality meat production emerges as a strategic direction. Breeding cattle in Spanish *dehesas*, using natural pastures and agricultural residues, not only improves meat quality but also contributes to addressing existing environmental sustainability issues [8].

This transition aligns with broader trends in the European Union, which recognize meat from native breeds as being of higher quality [9]. The EU currently promotes meat differentiation strategies, endorsing extensive livestock farming and certification systems based on origin and native breeds [10]. In Spain, the 100% Autochthonous Breed Logo serves as a quality mark, highlighting distinctive attributes, environmental relevance, and genetic heritage [11]. Therefore, amid the growing interest in sustainability, biodiversity, agroforestry systems and animal welfare, there is a unique opportunity to position Lidia meat differentially.

Despite the *Lidia* breed significance, an incomplete understanding of the quantitative and qualitative potential of its meat contrasts with the imperative to enhance its economic value. This is particularly crucial for female heifers unsuitable for reproduction and adult cows intended for the meat market, as highlighted by Buxadé [12]. Prior to slaughter, these animals undergo a four-month feed management period, incorporating concentrates and high-quality forage.

In the pursuit of differentiation, meeting consumer expectations for quality becomes essential [13]. Therefore, the aim of this study was to assess the carcass and meat quality of female *Lidia* cattle slaughtered at different ages, in order to deepen our understanding of the breed’s unique characteristics. This research intends to develop new insights into livestock adaptability, contributing to a broader knowledge of the diversity and sustainability of farming systems. Additionally, this study sought to provide robust empirical data supporting the potential differentiation and market positioning of *Lidia* meat, aligning with the increasing consumer criteria of sustainability, origin and quality.

## 2. Materials and Methods

### 2.1. Animals

This study included 300 carcasses from *Lidia* females collected from ten farms selected to represent existing diversity. Animals were slaughtered at three ages: I (Heifer I): 24–36 months old, II (Heifer II): 36–48 months old, III (Cull cows) > 48 months old [14]. Thirty animals were evaluated on each farm, ten for each slaughter age. 

Animals were selected from farms that follow the traditional production system of wild cattle. These breeding conditions are characterized by weaning at 7–8 months of age, open housing throughout the year with very low stocking rates, all-year grazing and supplementation with forage and concentrates in periods of grass scarcity (usually summer and winter). Four months prior to slaughter, the animals were supplemented with 3 kg of concentrate per day. A more detailed description of the production system can be found in the bibliography [15,16]. 

### 2.2. Carcass and Meat Quality Analyses

Before slaughter, animals were stunned using captive-bolt, complying with the current European regulations [17]. After slaughter, carcasses were suspended vertically using the Achilles method. At 1 h *post-mortem*, the carcasses were graded for conformation and fatness by trained staff using the EUROP system [18]. Conformation (CS) and fatness (FS) were scored on a 15-point scale (1—very bad conformation; 15—very good conformation; 1—very low fatness; 15—very high fatness, respectively), after which the carcasses were chilled and stored at 4 °C for approximately 24 h. Afterwards, the carcasses were split along the spinal column into two equal parts and the left-half carcass weights were recorded. 

The ultimate pH (pH_24_), meat and subcutaneous fat color and morphological measurements were assessed on the left side of each carcass. The pH_24_ was measured using a Hanna HI9025 portable pH-meter (Hanna Instruments, Laval, QC, Canada) with a penetrating glass electrode on the Longissimus thoracis muscle at the level of the 13th thoracic vertebra of the right side, at right angles to the sagittal plane surface. A Minolta 2600d spectrophotocolorimeter (Konika Minolta, Osaka, Japan), standardized against a white tile (L* = 97.78, a* = 0.19, b* = 1.84), with a D65 illuminant, an angle of 10° and an aperture size of 8 mm, was used to assess the color in CIELab* space of the m. *Rectus abdominis* and subcutaneous fat [19]. Readings for fat color were taken on subcutaneous fat covered with plastic food wrap (calibration was performed using the food wrap to maintain the integrity of the results). Three different locations were scanned and averaged for statistical analyses. Chroma (C*) and hue (h*) were calculated using the mathematical formula described by ISO regulations [19].

Standard measurements were then taken on the left half-carcass [20]. Length, depth of chest, hind-limb length and hind-limb perimeter of the carcasses were recorded. The left half-carcasses were then separated between the 5th and 6th thoracic vertebrae as forequarter and hindquarter. To assess the tissue composition, the 6th rib joint was removed by cutting the length of the bone at the limit of the m. *Serratus dorsalis* [21], which was weighed and dissected into lean, total fat (subcutaneous and intermuscular), bone and waste tissues (blood vessels, tendons).

For the instrumental and sensory analysis, a boneless section of the m. *Longissimus*, including the 12th and 13th thoracic vertebrae, and the m. *Longissimus lumborum* corresponding to the first two lumbar vertebrae, were removed from the left half-carcass. They were then packaged, aged at 4 °C for 21 days and frozen at −20 °C until the required evaluation and analysis. After this process, samples were removed from their bags and dried carefully with blotting paper. The pH and meat color (after 30 min blooming at ambient temperature) values were recorded as previously performed in the fresh carcass.

The water-holding capacity (WHC) was determined as thawing (TL), pressure (PL), drip (DL) and cooking (CL) losses. For TL evaluation, each sample was weighed frozen and thawed after a period of 24 h at 4 °C. The PL was determined following the Grau and Hamm method with the modifications described by Beriain et al. [22], and expressed as the percentage of juice expelled after the compression of 5 g meat samples with 2.25 kg applied for 5 min. The DL was determined by the method described by Honikel et al. [23]: a piece of meat (20 × 20 × 25 mm) devoid of connective tissue and fat was lightly blotted, weighed and suspended in a plastic bottle, ensuring that there was no contact with the walls, and placed in a refrigerator at 4 °C for 24 h [24]. The samples were then lightly blotted and reweighed. The DL was expressed as a percentage of initial weight. The CL was evaluated on meat samples of similar geometry, individually placed in plastic bags in a water bath at 90 °C until the internal temperature reached 70 °C (monitored by thermocouples inserted in the core) and cooled until it had fallen to 4 °C. They were taken from the bags, dried with a blotting paper and weighed. The CL was expressed as the percentage loss related to the initial weight. Then, for Warner–Bratzler shear force assessment, ten cuboid cores (1 cm^2^ × 2.5 cm) from each cooked steak were removed parallel to the predominant muscle fiber orientation and sheared using a Texture Analyzer (Model TA.XT-2, Texture Analyzer^®^ Stable Micro Systems, Surrey, UK) equipped with a Warner–Bratzler shear device (25 kg load cell) and a crosshead speed of 200 mm/min. The down stroke distance was 3 cm (the probe should cut the meat completely). The ten peak shear forces recorded per sub-sample were averaged.

### 2.3. Sensory Analysis

Sensory analysis of three animals per slaughter age were carried out [25]. A panel composed of sixteen panelists was recruited (8 men and 8 women), with an average age of 37 + 8 years old and previous experience in beef sensory evaluation. Panelists were selected and trained [26] in three sessions with the scale and attributes to utilize. According to Cittadini et al. [27], a control analysis of the trained panel was carried out using the Panel Analysis procedure of the XLSTAT-Sensory software version 2023.1.6.

A total of three sessions were carried out and each panelist tasted three samples of each slaughter age in a randomized order (a total number of nine samples were tasted by each panelist during the three sessions). No information about the experiment was provided before sessions. The sensory traits studied included: color, odor intensity, flavor intensity, juiciness, overall tenderness and overall acceptance [28]. Each variable was scored using a 1-to-9 category scale (low to high intensity, respectively).

This analysis was carried out at the facilities of the University of Córdoba (Spain) in a laboratory equipped in line with ISO standards [29]. Steak samples were thawed overnight prior to the test at 2–4 °C and then taken out, cut to a 2.5 cm thickness and placed in a room until they reached a temperature of 17–19 °C. Meat samples were cooked in an oven (Gastro M6, IberGastro, Lucena, Spain) preheated at 190 °C until an internal temperature of 70 °C was reached, monitored with type K thermocouples (HH374 Omega, Omega Engineering Inc., Norwalk, CT, USA). The samples were trimmed of any external connective tissue, cut into 2 cm side cubes, wrapped individually in coded aluminum foil (three-digit) and placed in hot plates at 50 °C until tasted, for no longer than 15 min. They were then presented together on white plates. The order of tasting was designed and explained to panelists trying to avoid the “first-order carry-over” effect [30]. Unsalted cookies and double-distilled deionized water were provided to clean the palate between samples.

### 2.4. Statistical Analyses

Firstly, a Kolmogorov–Smirnov test was used to verify normality, a Durbin–Watson test to detect the absence of autocorrelation of the residues and heteroscedasticity was evaluated using the White test [31]. The bivariate association between the carcass and meat traits was explored using Pearson correlations. A mixed model was used (XLSTAT version 2023.1.6) to examine the effect of the slaughter age on carcass and meat quality traits. The slaughter age was introduced as a fixed factor, while the farm was introduced as a random effect. A second linear mixed model was built to evaluate the specific effect of slaughter age on sensorial attributes. In this model, the session and the panelist were included as a random term. The pairwise differences between least-square means were assessed using the Student–Newman–Keuls (SNK) method. Differences were considered significant if *p* < 0.05.

## 3. Results

### 3.1. Carcass Traits

The left half-carcass weight (average 59.5 kg) significantly increased with age at slaughter (*p* < 0.001), and similar trends (*p* < 0.01) were observed for carcass measurements and the compactness index (Table 1). Carcasses were classified as O and O+ for conformation (straight to concave profiles, medium muscle development) and as 2 or 2+ for fatness (light fat cover, meat visible almost everywhere), with non-significant differences (*p* > 0.05) between age groups. The sarcopoietic potential of the *encastes* was evident in significant differences found in the hind-limb perimeter (*p* < 0.01) and conformation and fattening scores (*p* < 0.05). Carcasses of Lidia females resulted in short length (114 cm), shallowness (38.8 cm), short hind-limbs (70.8 cm) and a low compactness index (1.03), similar to others rustic bovine.

The left half-carcass weight showed a positive correlation with carcass length and depth of the chest, as well as with the hind-limb perimeter and carcass compactness. In contrast, it showed a weak correlation with fat cover classification, which also presented a weak correlation with carcass length and depth of the chest. The conformation score showed a weak and positive correlation with the hind-limb perimeter (Appendix A).

### 3.2. pH and Carcass Color

The average pH_24_ was 5.66 for all animals (Table 2); no significant differences were observed (*p* > 0.05) among age groups. Subcutaneous fat was characterized as dark and yellowish (L* = 59.8; b* = 23.5). An age group effect (*p* < 0.001) on a*, b* and C* was evident, with the highest values in older animals. The farm influenced (*p* < 0.05) b*, C*, and h*. Fat L* values decreased with age, though not reaching statistical significance. The m. *Rectus abdominis* appeared dark (L* = 38.5), reddish (a* = 15.9) and slightly yellowish (b* = 15.3), with age significantly influencing a*, b* and C*, although less so (*p* < 0.01) than in subcutaneous fat. The farm also influenced L* and h*.

The values of the colorimetric variables measured in the m. *Rectus abdominis* and subcutaneous fat were highly correlated. Fat L* values showed a weak negative correlation with the left half-carcass weight, carcass length, depth of the chest and the amount of fat in the rib. The hue angle measured in m. *Rectus abdominis* was weakly correlated with the half-carcass weight, carcass length, depth of the chest, carcass compactness and fat cover classification. Carcass length and depth of the chest were positively correlated with a*, b* and C* of the subcutaneous fat, and a* and C* of the m. *Rectus abdominis*. Fat b* and C* values were correlated weakly and negatively with the proportion of subcutaneous fat in the rib, and positively with the proportion of intramuscular fat (Appendix A).

### 3.3. 6th Rib Cut Dissection

Lean constituted the predominant tissue (58.6%), followed by bone (24.8%) and dissectible fat (14.3%), revealing a significant difference in fat percentage among age groups, with higher averages for cull cows (*p* < 0.05) (Table 3). The farm exerted a noteworthy effect (*p* < 0.001) on the tissue composition. Dissection losses were 3.02%, 2.95% and 2.69%, respectively, in groups I, II and III.

Positive correlations were observed between the left half-carcass weight, carcass length and depth of chest with the rub weight and its components, as well as with the conformation score and fat cover classification of the carcass (Appendix A).

### 3.4. Meat Traits

The meat from animals in Group I exhibited a significantly higher pH (*p* < 0.05) compared to the other age groups (Table 4). These differences were also noted in the b* values of the m. *Longissimus thoracis*. Regarding WHC, no significant differences (*p* > 0.05) were observed between age groups at slaughter, while drip losses were influenced (*p* < 0.01) by the farm. The WBSF with a mean value of 4.6 kg/cm^2^ remained unaffected by any of the considered factors.

No significant correlations were found between meat traits and any variable, except between colorimetric variables measured in the meat.

### 3.5. Sensory Analysis

The sensory profile analysis of aged meat is presented in Figure 1, where all attributes scored above 5 in all evaluated treatments. Statistical analyses revealed significant differences attributed to slaughter age, except for color and odor intensity, which did not differ between groups, maintaining an average score of 6.14 and 6.12, respectively.

The overall acceptance reached an average of 6.34, being significantly higher in cull cows (group III) than in heifers (I and II), with no significant differences between the two latter groups. The most highly rated attribute was flavor intensity, achieving an average of 6.21, while juiciness received the lowest rating at 5.95. Cull cows (group III) obtained significantly higher sensory scores (flavor intensity, overall tenderness and juiciness) than heifers (groups I and II), with no significant differences between these two groups.

## 4. Discussion

### 4.1. Carcass Traits

The carcass values observed are somewhat poor compared to what is found in meat breeds and other local breeds reared in the *dehesa*. As the *Lidia* breed has a remarkable temperament, it is difficult to include it in a feedlot system, hindering further conformational and fat development [15]. Regarding body size, the *Lidia* breed, classified as a small to medium-sized rustic breed, showed an age at slaughter (>24 months) higher than that commonly recorded in slaughterhouses (14–16 months) [32]. However, the left half-carcass weight (59.5 kg) was lower than that recorded in specialized and rustic breeds reared in their natural environment [33,34]. This was expected due to the lower weight of *Lidia* females compared to other native Spanish cattle breeds (300–400 kg vs. 400–600 kg), and it was anticipated that the weight of the left half-carcass would be greater in adult animals compared to young animals [35].

The morphometry of carcass *Lidia* females were similar to other rustic bovine breeds [36]. However, this was different from carcasses found in slaughterhouses, especially in the carcass compactness index [33,37,38,39]. Evaluated carcasses showed conformation and fatness scores lower than those recorded in other native and even in rustic breeds [33,40,41]. This difference in scores could be attributed to their degree of maturity, dietary restrictions, and low-fat content, which may be related to the fact that unimproved breeds have less subcutaneous fat than improved ones [42]. These results are in line with studies emphasizing that fatness scores depend on subcutaneous fat deposition, which can be more easily depleted during periods of low energy input [43].

While it is commonly known that increasing carcass weight improves conformation [44], in this study, a uniform carcass conformation score (*p* > 0.05) was found in all carcasses, in agreement with previous research [39]. The fatness score did not change with age at slaughter, which is consistent with other studies [45]. Based on conformation and fatness scores, the carcasses of *Lidia* females show certain similarities with other native Spanish breeds such as *Avileña*, *Retinta* or *Morucha* [4,39]. The variability and zoometric differentiation among the farms could account for the influence on hind-limb perimeter and carcass conformation [4].

### 4.2. pH and Carcass Color

The pH_24_ level significantly impacts meat quality, as a lower pH is often linked to enhanced tenderness and results in lighter meat. Ultimate pH values (5.7) in this study fall within the normal beef range (5.4–5.8) [46,47]. Our results are in line with studies on *Pajuna* steers [48] and young bulls from the *Retinta* breed [49], being lower than those for Limousin crossbred heifers fed with agro-industrial by-products [24]. 

In the CIELab* space, the subcutaneous fat of the carcass of Lidia females showed a lower value of lightness compared to carcass values of Spanish breeds (L* = 59.77 vs. L* = 66–71.2) and higher values of red (a* = 8.22, vs. a* = 2.8 to 3.4) and yellow (b* = 23.48; vs. b* = 7.6 to 18.1) [50,51,52]. The same was found when compared with adult steers (L* = 61.1–66.4; b* = 10.26 to 13.79). Extensive systems tend to be associated with carcasses with a lower fat lightness and higher yellowness index compared to intensive systems [53,54]. Carcasses from Lidia females showed a similar L* to other breeds, but with higher red (a* = 0.44 to 1.18) and yellow indices (b* = 6.92 to 9.88) [24,55]. The elevated a* value is attributed to increased myoglobin due to the higher physical activity of the *Lidia* breed [56], while the b* values may be the result of nutrition [24]. 

Concerning age at slaughter, results show higher a* and b* values in adult animals, confirming previous studies [57], partially in agreement with Galli et al. [58], and contrasting with Marenčić et al. [59], who found negligible effects on color parameters. Du Plesis and Hoffman [60] observed lower L* values and higher a* values at 30 months, indicating darker and redder meat.

Differences between farms in b* values from subcutaneous fat can be attributed to variability in pasture quantity and quality in their geographical areas [61], as well as differences in pH, according to Page et al. [62], who reported a negative correlation between pH and b* values.

The correlations found between the colorimetric variables and the carcass traits are in line with previous studies [63], highlighting that subcutaneous fat darkens as carcass weight and size increase. The positive correlations between carcass size and fat and *m. Rectus abdominis* a*, b* and C* values suggest a relationship between carcass size and composition and the color characteristics of fat and meat [63].

### 4.3. 6th Rib Cut Dissection

The 6th rib dissection method, proven to offer a more precise prediction of tissue composition, distinguishes our study from earlier research employing different rib cuts [36,55]. Tissue composition in animals from rustic breeds varies from more specialized beef breeds [64], aligning with the leaner meats typical of extensive systems. The dissection of the 6th rib revealed high bone content, a moderate fat percentage and medium to low lean percentage, resembling compositions of rustic and native breeds in extensive systems [33,36,38,65]. In contrast to the work of Vieira et al. [52] on adult steers, *Lidia* females exhibited higher bone values (24.8 vs. 13.5%) and lower fat values (14.3 vs. 27.3%). The percentage of lean and bone content remained consistent with the age at slaughter, contrary to that found by Albertí et al. [66], who reported decreases with increased carcass weight. Allometric indices of bone (~0.6–0.7) and lean (~1), alongside breed maturity, may influence young animals reaching bone and lean development similar to cows, according to [36]. The lean-to-bone relationship similarity (*p* > 0.05) between age groups suggested no significant changes.

The correlations found were as expected, including a negative correlation between intramuscular fat and CS [63,67]. However, the correlation found between FS and tissue composition variables differed from that found by Nogalski et al. [68], although the latter used the last three thoracic ribs, which are considered not as good predictors as the 6th rib [69].

### 4.4. Meat Traits

After the ageing period, the meat from *Lidia* females (pH = 5.7) was within the optimal range. The meat from group I presented the highest pH values, although there were no significant differences between age groups. Younger animals tend to have lower muscle glycogen contents and pH drop rates, making them more susceptible to stress, which can result in higher pH values [70].

Meat color is a crucial factor for consumers when purchasing [71]. *Lidia* female meat was darker and less red than that found in Spanish rustic [41,48,72] and specialized beef breeds [55], which are classified as very red [73]. Grazing, which leads to higher myoglobin content due to increased physical activity, contributed to the observed darker color [48,53,74]. Meat color did not follow the age-related evolution reported in some studies. The L* values were not affected by age at slaughter, contrary to previous findings [45,74]. However, a* values were lower than those found in the *Morucha* rustic breed [40], and b* values decreased with age [75]. Overall, the color variation was related to body size, fat content and muscle development [72].

The water holding capacity (WHC) is a crucial property for both the industry and consumers [76]. Thawing losses were similar to those in previous studies [49], and the ageing period did not significantly affect the WHC. Thawing losses were within the acceptable range, and meat was classified as juicy [77]. However, some variability in losses was observed, supporting the idea that leaner breeds have a lower WHC [38]. The influence of carcass weight on WHC was inconclusive, with no significant differences between age groups.

The Warner–Bratzler shear force (WBSF) values indicated that *Lidia*-female-cooked meat fell within acceptable limits for tenderness [63]. The values obtained aligned with those considered tender [78] or of intermediate tenderness [79]. No significant differences in WBSF were observed between age groups, which is possibly attributed to the higher fat content in cull cow carcasses [80]. While the effect of carcass weight on WBSF is not conclusive, some studies have reported significant changes with age at slaughter [45,81]. The specific activity of the calpain enzyme and collagen characteristics have been considered as sources of tenderness variation at different slaughter ages [82].

### 4.5. Sensory Analysis

The sensory profile analysis revealed significant differences attributed to slaughter age for flavor intensity, juiciness, overall tenderness and overall acceptance. Differences found are usually determined by diet [28,83] and ageing [84,85]. Except for color and odor, sensory attributes showed an increase with the age at slaughter, in contrast to some previous findings [58].

Meat color is influenced by various factors [86], and in this case presented satisfactory results for all evaluated slaughter ages. Odor intensity, often influenced by intramuscular fat content and fatty acid composition, received positive ratings [87,88]. Flavor intensity, a multifactorial trait [89,90,91], showed improvement with increasing age at slaughter. A higher age at slaughter might have generated a slightly higher amount of intramuscular fat deposits, which could cause the differences found between heifers and cows [92]. Likewise, it would be interesting to explore whether the fatty acid profile in the meat of cows versus heifers varied, which could be a possible justification for the differences found. Although numerically WHC variables were lower in cows, the results were not significant; however, the panel reported significant differences in meat juiciness. Juiciness scores were higher in cull cows, possibly due to a slightly higher fat content associated with age [93]. Overall tenderness improved with age, contrasting with some studies reporting an inverse relationship [94]. Overall acceptance differences correlated closely with flavor, juiciness and overall tenderness, emphasizing their collective importance in consumer perception. The exact hierarchy of importance among these attributes remains a subject of debate in the literature [95,96].

After evaluating the main carcass traits and meat quality of the *Lidia* breed, it would be of interest to subject it to assessment by untrained consumers, aiming for a deeper understanding of how *Lidia* breed meat could better align with the current market preferences.

## 5. Conclusions

This study reveals that the age at slaughter in *Lidia* females exerts a significant influence on carcass traits and meat quality. The increment of age at slaughter influenced the morphometry and a*, b* and C* color variables of carcass muscle and subcutaneous fat. The tissular composition of cull cows shows higher lean, fat and bone portions, and a higher percentage of fat. In contrast, instrumental variables were not influenced by the age groups studied. Important results were found on sensory variables; cull cows presented higher values in flavor intensity, juiciness, overall tenderness and overall acceptance. Although *Lidia* females are small to medium-sized, with a lower carcass weight compared to other breeds, the instrumental variables were acceptable, and their sensory characteristics suggest a positive potential for consumption. 

## Figures and Tables

**Figure 1 animals-14-00850-f001:**
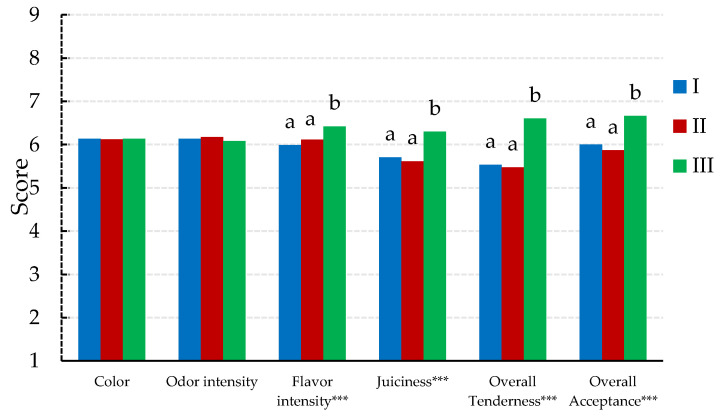
Mean values for the sensory attributes of female Lidia breed meat slaughtered at different ages. Means with different letters (a, b) are significantly different (SNK *p* < 0.05; *** *p* < 0.001). I = Heifer I (24–36 months); II = Heifer II (36–48 months); III = Cull cows (>48 months). A scale of categories from 1 to 9 (low to high intensity, respectively) was used.

**Table 1 animals-14-00850-t001:** Effect of age at slaughter (fixed effect) and farm (random effect) on carcass traits of female Lidia breed (mean ± SE ^1^).

Variable ^2^	Mean ± SE	Age at Slaughter ^3^				Farm
		I	II	III	*p*-Value	*p*-Value ^4^
LHCW (kg)	59.48 ± 0.93	53.41 ± 2.22 ^a^	56.77 ± 1.41 ^a^	68.27 ± 1.95 ^b^	<0.001	ns
CL (cm)	114.03 ± 0.46	108.59 ± 1.07 ^a^	110.75 ± 1.09 ^a^	122.76 ± 1.21 ^b^	<0.001	ns
DCh (cm)	38.80 ± 0.18	36.79 ± 0.43 ^a^	37.86 ± 0.27 ^b^	41.77 ± 0.38 ^c^	<0.001	ns
HL (cm)	70.84 ± 0.78	67.39 ± 1.80 ^a^	70.14 ± 1.14 ^a^	74.99 ± 1.59 ^b^	<0.01	ns
HP (cm)	78.48 ± 0.76	76.01 ± 1.84 ^a^	78.75 ± 1.16 ^ab^	80.67 ± 1.61 ^b^	<0.01	<0.01
CC (kg/cm)	1.03 ± 0.13	0.98 ± 2.22 ^a^	1.02 ± 2.22 ^a^	1.11 ± 2.22 ^b^	<0.01	ns
CS (1-15)	5.50 ± 0.04	5.52 ± 0.09	5.54 ± 0.06	5.44 ± 0.08	ns	<0.05
FS (1-15)	5.85 ± 0.15	5.86 ± 0.37	5.56 ± 0.23	6.14 ± 0.42	ns	<0.05

^1^: SE = standard error. ^2^: LHCW = left half-carcass weight, CL = carcass length, DCh = depth of chest, HL = hind-limb length, HP = hind-limb perimeter, CC = carcass compactness, CS = conformation score, CS = EUROP classification scales for conformation (from P − = 1 to 15 = E +), FS = fat cover classification (from 1 = low to 5 = very high). ^3^: Means with different letters (a, b) are significantly different (SNK *p* < 0.05). ^4^: ns = non-significant.

**Table 2 animals-14-00850-t002:** Effect of age at slaughter (fixed effect) and farm (random effect) on pH and color variables measured on subcutaneous fat and m. *Rectus abdominis* in carcasses of female Lidia breed (mean ± SE ^1^).

Variable ^2^	Mean ± SE	Age at Slaughter ^3^				Farm
		I	II	III	*p*-Value ^4^	*p*-Value ^4^
pH_24_	5.66 ± 0.03	5.67 ± 0.06	5.65 ± 0.04	5.66 ± 0.06	ns	<0.05
*Subcutaneous fat*						
L*	59.77 ± 0.63	62.17 ± 1.51	59.82 ± 0.97	57.33 ± 1.34	ns	ns
a*	8.22 ± 0.48	5.62 ± 1.15 ^a^	7.54 ± 0.73 ^a^	11.50 ± 1.02 ^b^	<0.001	ns
b*	23.48 ± 0.33	19.44 ± 1.74 ^a^	20.60 ± 1.12 ^a^	30.41 ± 1.54 ^b^	<0.001	<0.05
Chroma	25.06 ± 0.37	20.45 ± 1.97 ^a^	21.92 ± 1.27 ^a^	32.80 ± 1.74 ^b^	<0.001	<0.05
Hue angle	72.03 ± 0.80	74.64 ± 1.92	71.41 ± 1.23	70.06 ± 1.69	ns	<0.05
*Rectus abdominis*						
L*	38.46 ± 0.38	39.03 ± 0.89	38.12 ± 0.57	38.24 ± 0.79	ns	<0.01
a*	15.85 ± 0.37	14.94 ± 0.88 ^a^	14.69 ± 0.56 ^a^	17.90 ± 0.77 ^b^	<0.01	ns
b*	15.03 ± 0.38	13.58 ± 0.89 ^a^	15.13 ± 0.57 ^ab^	16.37 ± 0.79 ^b^	<0.01	ns
Chroma	21.93 ± 0.49	20.09 ± 1.16 ^a^	21.36 ± 0.74 ^a^	24.35 ± 1.02 ^b^	<0.01	ns
Hue angle	44.22 ± 0.53	46.42 ± 1.26	43.50 ± 0.81	42.75 ± 1.11	ns	<0.05

^1^: SEM = standard error. ^2^: L* = lightness, a* = redness, b* = yellowness. ^3^: Means with different letters (a, b) are significantly different (SNK *p* < 0.05). ^4^: ns = non-significant.

**Table 3 animals-14-00850-t003:** Effect of age at slaughter (fixed effect) and farm (random effect) on sixth rib joint dissection variables of female Lidia breed (mean ± SE ^1^).

Variable ^2^	Mean ± SE	Age at Slaughter ^3^				Farm
		I	II	III	*p*-Value ^4^	*p*-Value ^4^
Rib weight (g)	1991.76 ± 40.50	1885.90 ± 96.81 ^a^	1930.46 ± 66.14 ^a^	2158.92 ± 87.76 ^b^	<0.05	ns
SF thickness (cm)	0.20 ± 0.02	0.20 ± 0.04	0.22 ± 0.03	0.19 ± 0.04	ns	ns
Bone + waste (g)	483.05 ± 10.36	460.37 ± 24.65	480.79 ± 16.84	504.98 ± 22.34	ns	<0.01
Bone + waste (%) *	24.81 ± 0.40	24.78 ± 0.95	25.22 ± 0.65	24.43 ± 0.86	ns	<0.001
Lean (g)	1150.20 ± 27.16	1098.13 ± 64.98 ^a^	1110.85 ± 44.39 ^a^	1233.63 ± 58.91 ^b^	<0.05	ns
Lean (%) *	58.64 ± 0.55	59.12 ± 1.25	58.05 ± 0.85	58.77 ± 1.13	ns	<0.000
LT (g)	138.32 ± 3.57	136.96 ± 8.50	139.53 ± 5.80	138.46 ± 7.70	ns	<0.05
LT (%) **	12.02 ± 0.29	12.40 ± 0.40 ^b^	12.74 ± 0.27 ^b^	10.91 ± 0.36 ^a^	<0.05	ns
Fat (g)	312.66 ± 13.71	270.59 ± 26.37 ^a^	280.56 ± 18.01 ^a^	360.82 ± 23.90 ^b^	<0.05	<0.01
Fat (%) *	14.28 ± 0.55	13.31 ± 1.00 ^a^	13.26 ± 0.68 ^a^	16.25 ± 0.91 ^b^	<0.05	<0.000
SF (g)	48.88 ± 4.16	48.57 ± 9.86 ^ab^	35.48 ± 6.73 ^a^	58.60 ± 8.93 ^b^	<0.05	<0.01
SF (%) ***	14.75 ± 0.76	17.14 ± 1.81	12.67 ± 1.24	14.45 ± 1.64	ns	ns
IF (g)	266.77 ± 10.96	222.03 ± 26.37 ^a^	243.07 ± 18.01 ^a^	299.22 ± 23.90 ^b^	<0.05	<0.01
IF (%) ***	85.21 ± 0.76	82.86 ± 1.81	87.24 ± 1.24	85.53 ± 1.64	ns	ns

^1^: SE = standard error. ^2^: LT = m. *Longissimus thoracis*, SF = subcutaneous fat, IF = intermuscular fat, * respect to rib weight, ** respect to lean weight, *** respect to fat weight. ^3^: Means with different letters (a, b) are significantly different (SNK *p* < 0.05). ^4^: ns = non-significant.

**Table 4 animals-14-00850-t004:** Effect of age at slaughter (fixed effect) and farm (random effect) on meat traits from female Lidia breed aged 21 days (mean ± SE ^1^).

Variable ^2^	Mean ± SE	Age at Slaughter ^3^				Farm
		I	II	III	*p*-Value ^4^	*p*-Value ^4^
pH	5.73 ± 0.02	5.87 ± 0.06 ^b^	5.64 ± 0.04 ^a^	5.68 ± 0.06 ^a^	<0.01	<0.05
L*	25.49 ± 0.43	24.51 ± 1.02	26.13 ± 0.65	25.83 ± 0.90	ns	ns
a*	14.42 ± 0.33	14.28 ± 0.74	14.64 ± 0.47	13.90 ± 0.66	ns	ns
b*	12.86 ± 0.23	11.86 ± 0.54 ^a^	13.69 ± 0.35 ^b^	13.02 ± 0.48 ^ab^	<0.05	ns
Chroma	19.63 ± 0.34	18.77 ± 0.77	20.13 ± 0.49	19.08 ± 0.68	ns	ns
Hue angle	42.40 ± 0.64	40.38 ± 1.44	43.18 ± 0.93	42.86 ± 1.28	ns	ns
Thawing loss (%)	5.34 ± 0.26	5.45 ± 0.58	4.73 ± 0.37	4.37 ± 0.51	ns	ns
Drip loss (%)	0.97 ± 0.02	0.96 ± 0.13	1.11 ± 0.08	0.85 ± 0.12	ns	<0.01
Pressure loss (%)	8.90 ± 0.23	7.97 ± 0.51	9.21 ± 0.33	9.52 ± 0.45	ns	ns
Cooking loss (%)	22.16 ± 0.40	21.75 ± 0.97	23.12 ± 0.62	21.61 ± 0.88	ns	ns
WBSF (kg/cm^2^)	4.59 ± 0.16	4.07 ± 0.37	4.55 ± 0.24	5.14 ± 0.33	ns	ns

^1^: SE = standard error. ^2^: WBSF = Warner–Braztler shear force, L* = lightness, a* = redness, b* = yellowness. ^3^: Means with different letters (a, b) are significantly different (SNK *p* < 0.05). In the table ns refers to non-significant results. ^4^: ns = non-significant.

## Data Availability

The data presented in this study are available on request from the corresponding author. The data are not publicly available due to project IP rules.

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
