# Peer review of "Carcass and Meat Quality Traits in Female Lidia Cattle Slaughtered at Different Ages"

_animals, 2024, doi:10.3390/ani14060850_

Round 1

Reviewer 1 Report

Comments and Suggestions for Authors

The manuscript comprises the study of the effect of slaughter age on carcass traits and LD meat quality attributes from Lidia heifers and cows, reared and finished in an extensive system.

Before publication, certain aspects of the manuscript must be improved.

 Line 25

"To our knowledge, this is the first work evaluating carcass traits and technological and sensory quality of meat with a large sample of animals.”

I disagree with this sentence. For instance, the studies related to MSA (Meat Standard Australia) involved a large number of samples.

Line 114

It is necessary to include the illuminant, the observer angle, and the information of the calibration plates.

Line 133

Include the blooming temperature.

Line 168

The authors are asked to clarify whether they refer to overall or fiber tenderness, and to indicate how it was determined. Additionally, they are requested to clarify this in the rest of the manuscript.

Lines 147 and 173

Why different cooking methods were used for WBSF and sensory evaluation?

Line 191

How was the panelist considered in the model?

Line 117

It is necessary to establish a comparison when it is mentioned that fat is darker or more yellow.

Table 4

Authors are requested to include the word "loss", for example cooking loss.

Line 367

Analysis of the results is unclear. The researchers worked with a trained panel, not consumers. If these results are to be extrapolated to consumers, they should be clearly described and references should be included.

Line 375

The sensory attribute scores in this study were slightly higher than those obtained in other breeds……….”.

How was this comparison made? Did the same trained panel work in both trials?

Line 381

“color is influenced by various factors [85], and in this case was satisfactory and within the range of consumer acceptance”.

Please add references and explain the range.

Author Response

An attached document provides a point-by-point response to the reviewer's comments.

Reviewer 2 Report

Comments and Suggestions for Authors

Author Response

(The authors gave the same response as above.)

Round 2

Reviewer 2 Report

Comments and Suggestions for Authors

Thank you for your serious corrections.